# A Reinforcement Learning Method for a Hybrid Flow-Shop Scheduling Problem

**Wei Han, Fang Guo ***  **and Xichao Su**

Department of Airborne Vehicle Engineering, Naval Aviation University, Yantai 264001, China;
hanwei70cn@tom.com (W.H.); suxich@126.com (X.S.)

**\*** Correspondence: guofang575856@163.com

**Abstract:** The scheduling problems in mass production, manufacturing, assembly, synthesis, and transportation, as well as internet services, can partly be attributed to a hybrid flow-shop scheduling problem (HFSP). To solve the problem, a reinforcement learning (RL) method for HFSP is studied for the first time in this paper. HFSP is described and attributed to the Markov Decision Processes (MDP), for which the special states, actions, and reward function are designed. On this basis, the MDP framework is established. The Boltzmann exploration policy is adopted to trade-off the exploration and exploitation during choosing action in RL. Compared with the first-come-first-serve strategy that is frequently adopted when coding in most of the traditional intelligent algorithms, the rule in the RL method is first-come-first-choice, which is more conducive to achieving the global optimal solution. For validation, the RL method is utilized for scheduling in a metal processing workshop of an automobile engine factory. Then, the method is applied to the sortie scheduling of carrier aircraft in continuous dispatch. The results demonstrate that the machining and support scheduling obtained by this RL method are reasonable in result quality, real-time performance and complexity, indicating that this RL method is practical for HFSP.

**Keywords:** reinforcement learning; hybrid flow-shop scheduling problem; Markov decision processes; sortie scheduling of carrier aircraft

---

## 1. Introduction

The traditional flow shop scheduling problem can be described as $n$ workpieces to be processed on $m$ machines, each workpiece has to be machined in $m$ machines, and each machining stage must be worked on different machines. The order of $n$ workpieces processed on $m$ machines is the same, and the goal of the problem is to find the machining order of each workpiece on each machine [1]. HFSP is the integration of traditional flow shop scheduling and parallel machine scheduling [2,3]. With the characteristics of flow shop and parallel machine, HFSP is difficult to solve and even the two-stage HFSP is also an NP-hard (non-deterministic polynomial, NP) problem [4].

Based on the different type of parallel machines, HFSP can be divided into three categories: the same parallel machine HFSP, the uniform parallel machine HFSP, and the unrelated parallel machine HFSP [5]. In the same parallel machine HFSP, any workpiece has the same processing time on any parallel machine at each stage. The machining time of any workpiece on any parallel machine at each stage is inversely proportional to the processing speed of the machine in the uniform parallel machine HFSP. Explicitly, the working time of any workpiece on two parallel machines at each stage is not correlated with each other in the unrelated parallel machine HFSP, which is the focus of this paper.

HFSP has a strong engineering background and widely exists in chemical industry, metallurgy, textile, machinery, semiconductor, logistics, construction, papermaking and other fields. The research of HFSP is full of important academic significance and application value, for which HFSP has been widely

studied and applied up to now. To solve the HFSP, exact algorithms [6,7], heuristic algorithms [8], and intelligent algorithms [9–11] are mainly taken into account. Though the exact algorithms can get the optimal solution theoretically, its computation time is usually unacceptable, resulting in it only being suitable for solving small-scale HFSP. Heuristic algorithms, which are usually based on specific heuristic rules, can quickly obtain the solution of the problem; it is difficult, however, to guarantee the quality of the solution. In the past few years, an array of intelligent algorithms has been proposed and can be applied to solve HFSP effectively.

RL is an algorithmic method for solving sequential decision problems where an agent learns through trial and error interacting with its environment. As such, the agent is connected to the environment via perception and action such that the agent seeks to discover a mapping of system states to optimal agent actions. Having been a research hotspot in the field of machine learning in recent years, RL is widely adopted in industrial manufacturing [12], simulation [13], robot control [14], optimization and scheduling [15,16], game [17,18] and other fields, and achieves impressive performances.

Q-Learning [19], one of the most popular algorithms of RL, is used in this paper. In the Q-learning algorithm, the goal is to find a state–action pair value, which represents the long–term expected reward for each pair of state and action. The optimal state–action values for a system, proven to converge to the optimal state–action values, represent the optimal policy that an agent intends to learn.

To the best of the authors' knowledge, there has not been any reported research focused on HFSP with the RL method before. In this paper, the solution methods of HFSP and RL method are described in detail in the literature review section. The solving model is established in Section 3 and is attributed to the MDP framework in Section 4. In Section 5, the reinforcement learning for HFSP is verified on scheduling in the metal processing workshop of an automobile engine factory. Then, the method is utilized to the sortie scheduling of carrier aircraft in Section 6 and the concluding remarks are contained in the last section.

## 2. Literature Review

For solving HFSP, a lot of precious works have been done. On exact algorithms, Haouar et al. [20] proposed a branch and bound method (B&B) based on an improved search strategy. When the maximal number of workpieces reaches 1000, the deviation from the lower limit remains within 0.1%, but its performance remains to be improved for medium-scale problems with workpieces from 20 to 50. Tang et al. [21] studied real-time and wait-free HFSP in the background of steel production, established an integer programming model, and proposed an improved Lagrangian relaxation algorithm, which can achieve satisfactory optimization performance in fewer iterations, especially for large-scale problems.

Many scholars have devoted themselves to the research of heuristic algorithms for the rapidity of solution. To solve the two-stage unrelated parallel machine problem, Riane et al. [22] proposed a heuristic method based on dynamic programming and Low et al. [23] put forward a heuristic method based on an improved Johnson rule, which effectively solves two-stage HFSP with irrelevant parallel computers. In multi-stage HFSP, due to the complexity of the problem, the study of heuristic methods is rare. Hong et al. [24] proposed an improved fuzzy heuristic rule and studied the problem including fuzzy data. Ying et al. [25] researched the multi-stage HFSP with multiprocessors applying heuristic algorithm.

Recently, various intelligent algorithms have been being constantly proposed and effectively solve the HFSP, including the genetic algorithm (GA) [9], the simulated annealing algorithm [10], the tabu search algorithm [26], the ant colony optimization algorithm [27], the particle swarm optimization algorithm [28], the grey wolf optimization algorithm [29], the artificial bee colony algorithm [30], the artificial immune system (AIS) [31], the agent-based method [32], and so on. Xiao et al. [9] proposed a GA based on heuristic rules. To generate feasible scheduling, GA is used to allocate and sort machines in the first stage, while, in the second stage, FCFS is used to sort machines. Liu et al. [31] established a mixed-integer nonlinear programming model of HFSP with the minimum of makespan as the objective

function. Yu et al. [32] studied a multi-agent based hybrid flow shop dynamic scheduling system, in which the coordination mechanism between the various agents was designed.

Overall, a number of algorithms have been proposed to address HFSP. To the best of the authors' knowledge, nevertheless, there is still no existing research about the RL method for HFSP.

Being always the focus of academic research, RL has achieved remarkable development till now and the performance of RL algorithm has been continuously improved. Haarnoja et al. [33] proposed soft actor–critic and an off-policy actor–critic deep RL algorithm based on the maximum entropy reinforcement learning framework where the actor aims to maximize the expected reward while also maximizing entropy. Their method achieves state-of-the-art performance on a range of continuous control benchmark tasks. Haarnoja et al. [34] studied how maximum entropy policies, which are trained using soft Q-learning, can be applied to real-world robotic manipulation. Gao et al. [35] proposed a unified RL algorithm, Normalized Actor–Critic, that effectively normalizes the Q-function, reducing the Q-values of actions unseen in the demonstration data. Gu et al. [36] explored algorithms and representations to reduce the sample complexity of deep reinforcement learning for continuous control tasks. Nevertheless, existing literature about the RL for HFSP is still nowhere to be found.

The HFSP is researched in this paper and attributed to MDP for which the special states, actions and reward function are designed, on whose basis the Q-learning method is adopted to find the optimal policy.

## 3. Description of the Hybrid Flow-Shop Scheduling Problem

In this section, HFSP is introduced, and its standard model is established.

A typical HFSP is shown in Figure 1, there are S stages in total in the process of HFSP, and $m_1$, $m_2, \ldots, m_S$ machines are included in each stage, respectively. Every workpiece must be worked in all stages in a certain order. In each stage, however, any one machine can be selected for each workpiece. For instance, the sequence [*start*, $M_{12}$, $M_{24}$, $\ldots$, $M_{S3}$, stop] can be one of an order of a workpiece being machined.

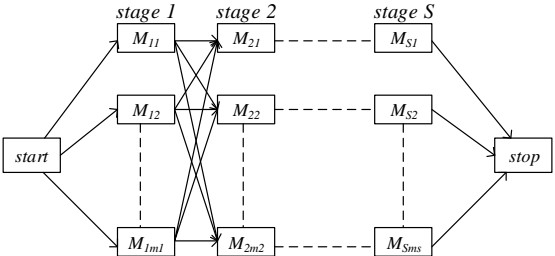

**Figure 1.** Flow chart of a typical HFSP.

In HFSP, several assumptions are usually made as follows: first, once a workpiece starts to be worked, it cannot be interrupted; second, a machine can only process one workpiece at a time; third, a workpiece can only be processed on one machine at a time; fourth, workpieces can be machined on any machine at each stage; fifth, there is unlimited storage capacity between any two stages (i.e., a workpiece can wait for any time between the two processes).

HFSP is divided into the determined initial sequence HFSP and the non-determined initial sequence HFSP in this paper, according to whether the initial sequence of the workpiece is deterministic or not. Knowing the processing time in each stage of the workpiece on each machine, the purpose of HFSP is to determine the distribution of each workpiece on the machines at each stage to minimize the maximum completion time for the former. For the latter, the ordering of all workpieces is necessary to define additionally.

$J_i(i = 1, 2, \ldots, n)$ indicates the workpieces needed to be worked, in which $n$ is the total number of workpieces. $m_j(j = 1, 2, \ldots, S)$ is the number of machines in each stage. $t_{i,j,l}$ is the machining time of the workpiece $J_i$ at stage $j$, machine $l$. $ct_{j,j+1}(j = 1, 2, \ldots, S-1)$ is the time cost on the way from

stage $j$ to stage $j + 1 (j = 1, 2, \ldots, S - 1)$. $AT_{i,j,l}$ is the arrival time of workpiece $J_i$ at stage $j$, machine $l$. Correspondingly, $ST_{i,j,l}$ and $ET_{i,j,l}$ indicate the starting and ending times of workpiece $J_i$ which is machined at stage $j$, machine $l$. $BT_{j,l}$ and $FT_{j,l}$ reveal the time when the machine $l$ at stage $j$ starts to work and stops working. The standard mathematical model of HFSP is as follows:

$$\text{minmax}\{ET_{i,j,l}\}\ i = 1, 2, \ldots, n; j = S; l = 1, 2, \ldots, m_S, \tag{1}$$

$$\text{s.t.} \sum_{l=1}^{m_j} y_{i,j,l} = 1\ i = 1, 2, \ldots, n; j = 1, 2, \ldots, S, \tag{2}$$

where $y_{i,j,l} = 1$ if workpiece $J_i$ is worked at stage $j$, machine $l$, else $y_{i,j,l} = 0$

$$ST_{i,j,l} = \max\{AT_{i,j,l}, FT_{j,l}\}\ i = 1, 2, \ldots, n; j = 1, 2, \ldots, S; l = 1, 2, \ldots, m_j, \tag{3}$$

$$ET_{i,j,l} = ST_{i,j,l} + t_{i,j,l}\ i = 1, 2, \ldots, n; j = 1, 2, \ldots, S; l = 1, 2, \ldots, m_j, \tag{4}$$

$$AT_{i,j+1,l} = ET_{i,j,l} + ct_{j,j+1}\ i = 1, 2, \ldots, n; j = 1, 2, \ldots, S - 1, \tag{5}$$

$$BT_{j,l} = ST_{i,j,l}\ j = 1, 2, \ldots, S; l = 1, 2, \ldots, m_j, \tag{6}$$

if workpiece $J_i$ is the first one machined at stage $j$, machine $l$ and without interruption.

$$FT_{j,l} = BT_{j,l} + t_{i,j,l}, \tag{7}$$

if workpiece $J_i$ is worked at stage $j$, machine $l$.

Equation (1) is the objective function, minimizing the maximum ending time of all workpieces in the last stage S. Equation (2) ensures that there is only one workpiece at any machine in any stage. Equation (3) clears the time limitation of arrival and start. Equation (4) shows the relationship between the completion time and the start time of the process at the same stage. Equation (5) calculates the arrival time on the next stage utilizing the end time of the previous stage. Equations (6) and (7) reveal the time when the machine $l$ at stage $j$ starts to work and stops working.

## 4. MDP Framework for HFSP

In this section, MDP is introduced to which HFSP is attributed. Firstly, the MDP framework is described in detail, and the states and actions of MDP framework for HFSP are determined in Section 4.2. The exploration and exploitation policy, an improved $\varepsilon$-greedy policy and Boltzmann exploration policy included, is analyzed in Section 4.3. The reward function based on machining time of each workpiece is designed in Section 4.4. The RL process for HFSP is contained in the last section.

### 4.1. Description of MDP

Reinforcement learning is usually described with an MDP. In HFSP, obviously, the location of the workpiece in the next stage is only related to the current position, but not to the previous location, which is Markov property, manifesting HFSP can be attributed to a MDP.

An MDP is a tuple $(S, A, P, R)$, where $S$ denotes a set of states, $A$ is a set of actions, $P : S \times A \mapsto [0, 1)$ is the state transition distribution upon taking action $a$ in state $s$, and $R : S \times A \times S' \mapsto R$ is the reward after taking action $a$ and transferring to state $s'$ in state $s$.

An agent in reinforcement learning learns the policy $\pi$ by continuously interacting with the environment over a number of time steps and getting environmental feedback, at each time step $t$, the agent selects the action $a$ at state $s_t$ and transfers to the next state $s_{t+1}$ from the policy $\pi(s)$. A policy $\pi : S \times A \mapsto R_+$ is a mapping from state–action pair to the probability of taking action $a$ in state $s$, so $\sum_{a \in A} \pi(s, a) = 1\ (\forall s \in S)$.

The Q-learning algorithm is one of the most illustrious algorithms in reinforcement learning. The action-value, or $Q(s, a)$, of a particular state under policy $\pi$, is:

$$Q(s, a) = E\left(\sum_{k=0}^{\infty} \gamma^k r_{t+k+1} \mid s_0 = s, a_0 = a, \pi\right), \tag{8}$$

where $r_{t+1}(s, a)$ is the reward function in time step $t$, and $\gamma$ is the discount factor.

The Bellman operator and the optimal Bellman operator for policy is defined in literature [37] as Equations (9) and (10):

$$T^{\pi} Q(s, a) = \underset{s', r, a'}{E} (r(s, a) + \gamma Q(s', a')), \tag{9}$$

$$T^* Q(s, a) = \underset{s', r}{E} (r(s, a) + \gamma \max_{a'} Q(s', a')), \tag{10}$$

where the expectation is over next state $s' \sim P(\cdot, s, a)$, the reward $r(s, a)$, and the action $a'$ is from policy $\pi(s')$. Both the Q-value function and the optimal Q-value function are the fixed points of corresponding Bellman equations. Bertsekas [38] proved the uniqueness of the fixed point of each operator and the convergence of value iteration. The iteration equation of Q-value is represented as follows [39]:

$$Q(s, a) = Q(s, a) + \alpha(r + \gamma \max_{a'} Q(s', a') - Q(s, a)), \tag{11}$$

where $\alpha$ is the learning rate.

### 4.2. Abstraction of State and Action for HFSP

States are the tuple $(stage, workpiece)$ in the MDP framework for HFSP, in which every workpiece chooses action and transfers to the next stage. In addition, the start stage is taken as an initial stage, where all workpieces lie before starting to be worked. When all workpieces in a stage transfer to the next stage, the state transfers to the next state. Actions are the machines in each stage. Take *machine* 1 at *stage 1* in Figure 1 as an example. There are $m_2$ machines in next stage, so any workpieces machined on *machine* 1 at *stage* 1 have $m_2$ actions to choose.

### 4.3. Exploration and Exploitation

The agent needs to continuously interact with the environment in Q-learning. The agent selecting the correct action based on the perceived external information determines whether the interaction is effective or not. When selecting an action, the action maximizing the Q-value function should be selected in each state to obtain as many rewards as possible; this is exploitation. On the other hand, the agent should explore the better actions to obtain the optimal Q-value so as not to fall into the local optimal value. To solve the problem, $\varepsilon$-greedy policy is often adopted, where the agent, given a coefficient $\varepsilon \in [0, 1]$, with the probability $1 - \varepsilon$, selects the action with the largest Q-value and randomly selects an action to execute with the left $\varepsilon$ probability. Undoubtedly, $\varepsilon$-greedy policy increases the probability to choose the better action at the beginning of learning. However, it affects the learning efficiency with an array of explorations when the policy is close to the optimal one in the later stage of study. Two solutions are usually available to solve the flaw of $\varepsilon$-greedy policy.

#### 4.3.1. Improved $\varepsilon$-Greedy Policy

The central blemish of $\varepsilon$-greedy policy is its high exploration at the end of study, for which an iteration equation is put forward in this paper, as shown in Equation (12), where the coefficient $\varepsilon$ gradually decreases with the episode of learning:

$$\varepsilon = \varepsilon_0 - \beta e. \tag{12}$$

This can be called improved $\varepsilon$-greedy policy where $\beta$ is a small decimal, and $e$ is the number of iterations. Note that $\varepsilon$ is not less than 0 must be guaranteed.

### 4.3.2. Boltzmann Exploration Policy

Boltzmann exploration policy is to determine the probability that each action is selected with a random distribution function. Given random temperature coefficient $T(>1)$, the probability of the $i^{th}$ action being selected in time step $t$ at state is:

$$p(s_t, a_{it}) = \frac{e^{Q(s_t, a_{it})/T}}{\sum\limits_{i}^{N} e^{Q(s_t, a_{it})/T}},$$
(13)

where $N$ is the total number of actions.

The temperature coefficient $T$ is high and Q-value is relatively small at the beginning of the learning, resulting in all the action to be chosen with nearly equal probabilities, which is beneficial for the agent to explore the actions with non-optimal Q-value. As the learning progresses, the temperature coefficient $T$ decreases gradually, and the probability changes with $Q(s, a)$, the probability of adopting random actions decrease, which is instrumental in selecting the optimal action with the largest Q-value. In the later stage of learning, the temperature parameter $T$ tends to 0, $Q(s, a)$ is the largest, and the corresponding action is selected with the largest probability, and the action with the largest Q-value is selected each time, which means the policy change to greedy policy.

The iteration of temperature coefficient is usually achieved by three policies:

$$T = \lambda^e T_0,$$
(14)

$$T = \frac{T_0}{\log(e + e_0)},$$
(15)

$$T = \frac{T_0}{e + e_0},$$
(16)

where $\lambda(\in (0, 1))$ is the cooling parameter, often set as a decimal close to 1. $e$ is the times of iteration and $e_0$ is a positive constant. $T_0$ is the initial value of temperature coefficient $T$; if set too small, it will cause the local minimum value of the algorithm; on the contrary, the calculation of the algorithm will be increased with a large $T_0$. Equation (14) is an equivalent cooling policy, Equation (15) is logarithmic cooling strategy, and Equation (16) is a fast cooling policy.

### 4.4. Reward Function Representation Based on Machining Time

Distinguishing an optimal policy from other ones is the reason why action-value, or $Q(s, a)$, is adopted, whose ability to express has a strong dependence on the representation of the reward function. In addition, to achieve plummy learning results and improve learning convergence speed, reward function representation is pivotal.

The ultimate goal of Q-learning is to maximize the cumulative reward, and the objective function is to minimize the machining time of all workpieces in this paper. Namely, the reward function is negatively correlated to machining time. For uniform representation, at the same time, a linear reward function is proposed, so the reward function in reinforcement learning for HFSP is defined as what follows:

$$r(s, a) = -\omega \times c\_t_{i,j,l} + b,$$
(17)

$$c\_t_{i,j,l} = FT'_{j,l} - BT_{j,l},$$
(18)

where $c\_t_{i,j,l}$ is the waiting time of workpiece $J_i$ before finishing being machined on machine $l$ at stage $j$ if workpiece $J_i$ chooses action $a$ in stage $j - 1$ and transfers to machine $l$ at stage $j$. $FT'_{j,l}$ is the updated finishing time after machine $l$ at stage $j$ being selected by action $a$:

$$FT'_{j,l} = FT_{j,l} + t_{i,j,l}. \tag{19}$$

Note that $BT_{j,l}$ is the time when machine $l$ begins to work, which is not necessarily equal to $ST_{i,j,l}$. $\omega$ and $b$ is a positive constant, making the reward function negatively correlated with $c\_t_{i,j,l}$. Generally, it takes two to five to distinguish the reward function of good action from others.

### 4.5. Reinforcement Learning Process for HFSP

For the deterministic initial sequence HFSP, its pseudo code of solution with reinforcement learning method is shown in Algorithm 1, and the corresponding flow chart is illustrated in Figure 2.

For the non-deterministic initial sequence HFSP, it is necessary to choose the initial sequence before executing the below pseudo code (i.e., Algorithm 1).

---

**Algorithm 1.** The Reinforcement Learning Method for HFSP

---

**Require:** discount factor $\gamma$, learning parameter $\alpha$

  initialize Q arbitrarily (e.g. $Q(s, a = 0; \forall s \in S, \forall a \in A)$

  **for each** episode **do**

    $s$ is initialized as the initial state $ST_{i,j,l}$, $ET_{i,j,l}$, $BT_{j,l}$, $FT_{j,l}$ are initialized as 0, $AT_{i,j,l}$ is initialized with the initial sequence.

    **repeat**

      **for each** state **do**

        **repeat**

          $AT_{i,j+1,k} = ET_{i,j,k}$ $(j = 1, 2, \ldots, S-1)$

          **for each** workpiece $J_i$ **do**

            **repeat**

              choose an action $a \in A(s)$ based on Q and an exploration strategy

              perform action $a$

              observe the new state $s'$ and receive reward $r$

              $Q(s, a) := Q(s, a) + \alpha(r + \gamma \max_a Q(s'a) - Q(s, a))$

              $i = i + 1$

            **until** all workpieces transfer to $s'$

          sort the workpiece with the ending time of machine with ascending

        **until** $s'$ is a goal state

      **until** episode is the last episode

    end

---

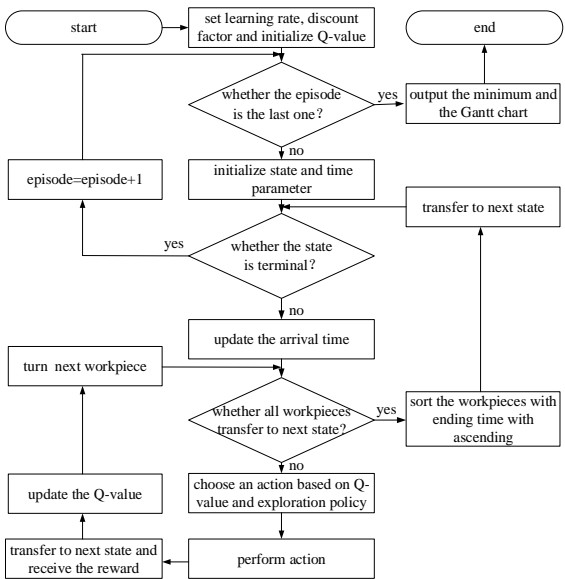

**Figure 2.** The flow chart of the reinforcement learning for HFSP.

## 5. Case Validation

In this section, an example of scheduling in the metal processing workshop of an automobile engine factory, a non-deterministic initial sequence HFSP, is utilized to verify the proposed algorithm. In addition, the reasonability of the parameters is verified and the complexity is analyzed.

### 5.1. Case Description

Compared to deterministic sequence HFSP, non-deterministic sequence HFSP poses a greater challenge to reinforcement learning for its stochastic initial arrival sequence. Only the non-deterministic sequence HFSP is validated, therefore, in this paper.

In this case, there are 12 workpieces, tagged with J1 to J12, needing to be processed. Each workpiece has three processes: lathing, planning, and grinding. Three lathes, two planers and four grinders (each labeled with *Mx*) are equipped. Each machine has different processing times for each different workpieces, which is shown in Table 1 (data from [40]). The time of scheduling between two machines in different stages is ignored in this case.

**Table 1.** Processing time of each workpiece on each machine.

| Workpiece | | J1 | J2 | J3 | J4 | J5 | J6 | J7 | J8 | J9 | J10 | J11 | J12 |
|---|---|---|---|---|---|---|---|---|---|---|---|---|---|
| | *M1* | 2 | 4 | 6 | 4 | 4 | 6 | 5 | 3 | 2 | 3 | 5 | 6 |
| lathing | *M2* | 2 | 5 | 5 | 3 | 5 | 5 | 2 | 5 | 5 | 6 | 2 | 5 |
| | *M3* | 3 | 4 | 4 | 4 | 3 | 4 | 4 | 4 | 4 | 4 | 4 | 4 |
| planing | *M1* | 4 | 3 | 4 | 6 | 3 | 2 | 4 | 7 | 1 | 3 | 3 | 5 |
| | *M2* | 5 | 4 | 2 | 5 | 1 | 3 | 6 | 5 | 2 | 4 | 5 | 4 |
| | *M1* | 2 | 3 | 3 | 3 | 3 | 4 | 3 | 3 | 7 | 4 | 6 | 3 |
| grinding | *M2* | 3 | 4 | 4 | 6 | 4 | 3 | 4 | 3 | 8 | 8 | 7 | 4 |
| | *M3* | 2 | 5 | 2 | 5 | 6 | 9 | 3 | 6 | 6 | 6 | 6 | 7 |
| | *M4* | 3 | 4 | 5 | 8 | 5 | 5 | 5 | 4 | 5 | 7 | 5 | 5 |

### 5.2. Parameters Setting

The parameters in the reinforcement learning method for HFSP are discussed in this section.

When trading-off the exploitation and exploration, the Boltzmann exploration policy with fast cooling policy (i.e., Equation (14)) is utilized. The initial temperature coefficient $T_0$ is set as 500, which

is relative to the different episodes of simulation of each initial sequence. The parameters of Q-learning are set as follows: the learning rate $\alpha = 0.1$ and discount factor $\gamma = 0.9$.

In Equation (17), to discuss how the coefficients $\omega$ in reward function affect the scheduling result (the minimum scheduling time), a random initial sequence [1–12] is tested when $\omega$ is set as the integers from 1 to 6, respectively, during which the value of $b$ is set as 200 to satisfy that the reward function $r(s, a)$ is not less than 0. The minimal scheduling time of 200 episodes' simulation, which goes with the coefficient $\omega$, is illustrated in Figure 3. From the result shown in Figure 3, the reward function can be set as $r(s, a) = -4c\_t_{i,j,l} + 200$.

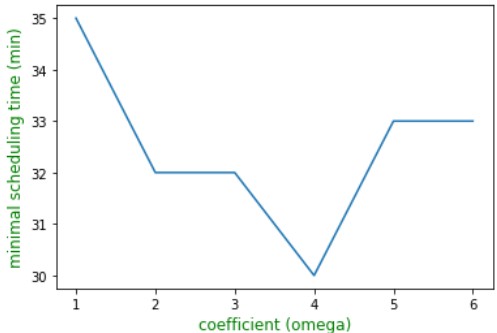

**Figure 3.** The effect of coefficient $\omega$ on minimal scheduling time for a random initial sequence.

### 5.3. Case Results

On the basis of Section 4.5, 100 initial sequences are generated randomly. For each initial sequence, 200 episodes' simulation are conducted, of which the minimal scheduling time is saved. The scheduling time of 100 initial sequences is minimized as the optimal scheduling time, and the corresponding initial sequence and computing time are also recorded.

The computing environment is 2.20 GHz, 8.00 GB, Win 10, Dell PC (City, US State abbrev. if applicable, Country), in which the simulation is executed 10 times in Spyder 3.3.3 in Anaconda 3 (March 2019) (64-bit) (Manufacturer, City, US State abbrev. if applicable, Country). The optimal scheduling time of 10 executing is listed in Table 2 where one result calculated by AIS [31] and 10 results by GA [40] are listed for comparison as well.

**Table 2.** The comparison of optimal results for 10 times (min).

| Times | 1 | 2 | 3 | 4 | 5 | 6 | 7 | 8 | 9 | 10 |
|-------|----|----|----|----|----|----|----|----|----|----|
| AIS | 27 | | | | | | | | | |
| GA | 29 | 30 | 29 | 29 | 29 | 31 | 29 | 29 | 29 | 30 |
| RL | 27 | 28 | 28 | 28 | 27 | 28 | 28 | 27 | 28 | 28 |

As Table 2 illustrates, the RL method can find a better solution of 27 min in contrast with GA. However, the result from RL is not better than the result from AIS, which may be due to the non-deterministic sequence of HFSP and the low stochastic sequences in the RL method.

The initial sequence corresponding to one of the optimal values in the RL method is [2–12] (the result of the 8th execution), of which the Gantt chart is shown in Figure 4.

In Figure 4, the *y*-label is the location where the workpieces are worked in each stage, for example, 'S3_1' indicates the first machine of the third stage or grinding stage. The annotation in the chart manifests the stage of machining and the tag of workpiece (e.g., '2-J1' demonstrates the workpiece with tag J1 being worked in the second stage or planing stage). The computing time is 20.5 s.

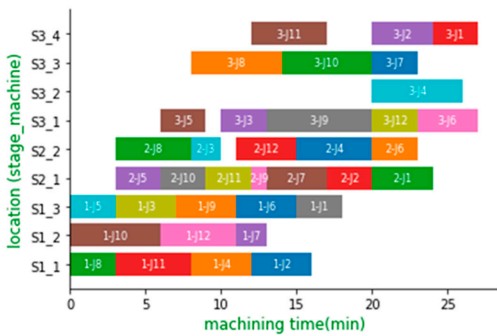

**Figure 4.** Gantt chart of one of the optimal scheduling of workpieces.

*5.4. Results Discussion*

The complexity of three methods are analyzed, and the result is discussed in this section.

The complexity of the RL method is O($|IS||E||S||n|$), where $IS$ is the number of initial sequences, $E$ is the number of episodes and $S$, $n$ is the number of stages and workpieces. The complexity of GA is O($|N||EG||S||n|$), where $N$ is population size, $EG$ is evolutionary generations and $S$, $n$ is the same as those in RL. The complexity of AIS is O($|N||EG||S||n|$), the meaning of the parameters is the same as those in GA, but different values of $N$, $EG$ are set in the scheduling problem. The complexity, the optimal scheduling time, and the computing time are analyzed and listed in Table 3.

**Table 3.** The complexity and optimal scheduling time of three methods.

| Method | AIS [31] | GA [40] | RL |
|---|---|---|---|
| parameter values | $N = 40$, $EG = 100$, $S = 3$, $n = 12$ | $N = 30$, $EG = 100$, $S = 3$, $n = 12$ | $IS = 100$, $E = 200$, $S = 3$, $n = 12$ |
| complexity | O($|N||EG||S||n|$) | O($|N||EG||S||n|$) | O($|IS||E||S||n|$) |
| optimal scheduling time | 27 min | 29 min | 27 min |
| computing time | – | – | 19 to 21 s |

Table 3 shows that the complexity of three methods is in the same order of magnitude in terms of the complexity. In terms of the optimal scheduling time, however, the AIS and RL methods are better than the GA method. The computing time is not given in AIS and GA but usually tens of seconds, the 19 to 21 s of computing time of the RL method can meet the actual needs of production.

Obviously, the non-deterministic initial sequences increase the complexity of the RL method and prominently influence the result quality. In addition, the results from the RL method, of course, remain to be improved, such as the effects of the parameters on the result, including $\gamma$, $\alpha$, $\omega$, $b$, and $T_0$, are not fully explored. Furthermore, it is reasonable to reason that the complexity of the RL method will be prominently superior to the AIS and GA method for the deterministic initial sequence scheduling problem.

Stated thus, though remaining to be improved, the RL method precedes GA in terms of the optimal scheduling time for the non-deterministic initial sequence scheduling problem, and it has tremendous potential to solve the deterministic initial sequence scheduling problem with appropriate parameter setting.

## 6. Application

In this section, the theory of reinforcement learning method for HFSP is applied to the sortie scheduling of carrier aircraft in continuous dispatch.

*6.1. Description of Carrier Aircraft Deck Operations Problem*

Multi-station support is still widely used in carrier aircraft deck operations today. Efficient carrier aircraft deck operations play an important role in improving aircraft carrier combat capability. There are many tasks of deck aviation support for carrier aircraft, which involve the transfer of carrier

aircraft among technical stations and the allocation of various resources on deck. According to the content of support, the technical station is divided into *detection and maintenance* stage, *refuel* stage, *rearm* stage, and *ejection* stage [41], each of which is equipped with four stations in this paper (i.e., station with certain support personnel and resources) to provide support services, and the support time of each station is distinct due to the different capabilities of personnel. The support time of each station for a similar carrier aircraft is shown in Table 4, which is from the supporting experience. Since different stations are located at different locations, it takes a certain amount of time to schedule between two stations. Transferring time between *detection and maintenance* stage and *refuel* stage, and *refuel* stage and *rearm* stage both obey Gauss distribution $N(2, 0.1)$; *rearm* stage and *ejection* stage obey Gauss distribution $N(4, 0.2)$.

**Table 4.** The support time of each station for a similar carrier aircraft (minute).

| Stage | Detection and Maintenance | Refuel | Rearm | Ejection |
|---|---|---|---|---|
| station 1 | 11 | 15 | 20 | 2 |
| station 2 | 9 | 13 | 15 | 2 |
| station 3 | 10 | 12 | 17 | 3 |
| station 4 | 13 | 16 | 13 | 3 |

Regarding the stations as machines, the carrier aircrafts needing to be supported as similar workpieces, sortie scheduling of carrier aircraft, evidently, can be attributed to HFSP considering the transferring time between stages. The flow chart of carrier aircraft deck operations is shown in Figure 5.

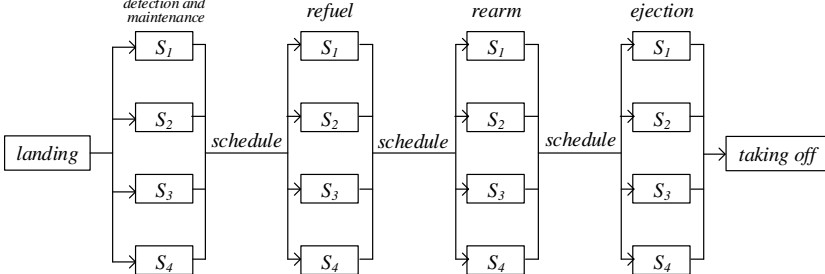

**Figure 5.** Flow chart of carrier aircraft aviation support scheduling.

*6.2. Simulation Results*

The Boltzmann exploration policy is adopted when choosing action, and the fast cooling policy (i.e., Equation (14)) is selected when iterating the temperature coefficient $T$. According to the principle in Section 5.2, the relative parameters are set as follows: $T_0 = 900$, $\gamma = 0.9$, $\alpha = 0.1$, and $r(s, a) = -2c\_t_{i,j,l} + 150$.

Suppose 20 carrier aircraft (labeled with 1 to 20) with similar conditions come to the *detection and maintenance* stage every 2 min, and the arrival time of the first one is 0. Thus, the arrival time to the first stage and the initial sequence of all aircraft is deterministic.

One thousand episodes of simulation are conducted in the same computing environment with Section 4.3, costing 2.03 s, the curve of maximal ending time of carrier aircraft support with an episode is shown in Figure 6a, and the former 200 episodes are shown in Figure 6b.

As illustrated in Figure 6, the curve converges from around the 90th episode, and the mean and variance of support time of the last 900 episodes are 135.3 min and 0.26, respectively, which prove the robustness of this method. To make clear the distribution of each carrier aircraft on the station at each stage, the Gantt chart of carrier aircraft scheduling of the 1000th episode is illustrated in Figure 7.

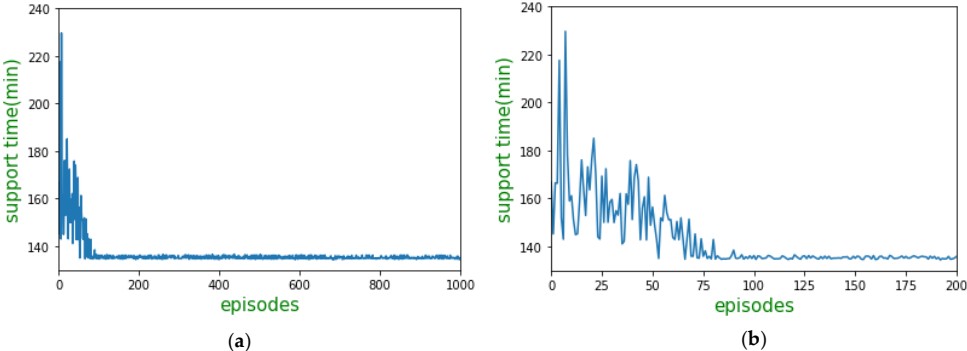

**Figure 6.** The curve of support time of carrier aircrafts with episodes. (**a**) Description of the curve of maximal support time during one thousand episodes; (**b**) Illustration of how the maximal support time goes with the episode in detail in the former 200 episodes.

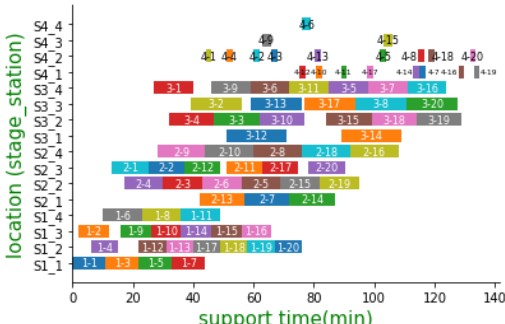

**Figure 7.** Gantt chart of optimal scheduling of aircrafts.

In Figure 7, the *y*-label is the location of carrier aircraft, for instance, 'S2_1' indicates the first station of the second stage or *refuel* stage. The number in the chart manifests the stage of security and the tag of aircraft (e.g., '2-1' demonstrates that the aircraft with tag 1 is in the second stage or *refuel* stage). The terminal support time is 134.9 min this episode.

For deterministic initial sequence HFSP, the main factors affecting the convergence speed and the result quality are the setting of parameters. Moreover, the stochastic transferring time between two stages also has some influence on the optimal scheduling time.

## 7. Conclusions

HFSP based on reinforcement learning is addressed in this paper. The literature investigation shows the universality of HFSP and its wide studies in various research methods, among which the reinforcement learning method is not included.

Firstly, HFSP and its model are introduced, and divided into deterministic initial sequence HFSP and non-deterministic initial sequence HFSP. HFSP is described into MDP, with the stage–workpiece pairs considered as states and the machines in the next state as actions. The reward function is set to be related to the processing time of the workpiece. To trade-off the exploration and exploitation, $\varepsilon$-greedy policy, improved $\varepsilon$-greedy policy, and Boltzmann policy are introduced. The reinforcement learning method for HFSP is validated in the case of scheduling in the metal processing workshop of an automobile engine factory; then, it is applied to the sortie scheduling of carrier aircraft.

The main contributions of this paper is the first application of a reinforcement learning method to HFSP, and the corresponding model is then established. The results obtained by this method, of course, are not necessarily the optimal ones, but they can provide the relative people with some reference for HFSP compared with manual scheduling and some intelligent algorithms scheduling, and this

method achieves satisfactory real-time performance in deterministic initial sequence HFSP through the application in this paper.

In the future, the effects of more relative parameters on the result of RL method will be analyzed. Moreover, combining intelligent algorithms with RL to solve non-deterministic initial sequence HFSP and improve the performance of deterministic initial sequence HFSP may be considered. Then, the different support time of each carrier aircraft and scheduling time between two stages will be fully investigated for constructing a more precise model of sortie scheduling of carrier aircraft. Finally, a repository of the sortie scheduling of carrier aircraft with different numbers of carrier aircrafts is about to be established based on the RL method to provide deck dispatchers with some instructive guidance.

**Author Contributions:** Conceptualization, F.G.; Methodology, W.H.; Software, F.G.; Validation, W.H. and X.S.; Formal analysis, F.G.; Resources, X.S.; Writing—original draft preparation, F.G.; Writing—review and editing, F.G.; Supervision, W.H.

**Funding:** This research received no external funding.

**Acknowledgments:** We are deeply grateful for the constructive guidance provided by the review experts.

**Conflicts of Interest:** The authors declare no conflict of interest.

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
