# Peer review of "A Reinforcement Learning Method for a Hybrid Flow-Shop Scheduling Problem"

_algorithms, doi:10.3390/a12110222_

Round 1

Reviewer 1 Report

This paper presents a reinforcement learning method which is called hybrid flow-shop scheduling problem (HFSP), which can be used for solving the scheduling problems in mass production, manufacturing, assembly, synthesis, and transportation, as well as internet services. The HFSP is based on the Markov Decision Processes (MDP) and the Boltzmann exploration policy is adopted to trade-off the exploration and exploitation during choosing action in reinforcement learning. The authors describe with a lot of details their initial thoughts and the reasons why this study is important. In addition, the paper includes results that might be useful for those interesting in solving the scheduling problem. Moreover, based on the provided results, it is demonstrated that the machining and support scheduling obtained by this reinforcement learning method is practical for HFSP. 

At this point, I would like to make some recommendation to the authors in order to improve their paper and make it ready for a journal publication.

1. Introduction: It is too short and it does not even introduce the problem and the method that was followed to solve the problem. Please consider revising this section.

2. There is no Literature Review section in this paper (or there is. some lit review in the Introduction section), which in most cases is Section 2. Instead, Section 2 describes the HFSP. I would recommend including a literature review section with prior work that has been conducted towards traffic light period configuration.

3. Current Section 2 should become Section 3 and so on...

4. Section 3 is quite informative but I would like to see a table with notations at the beginning. This would make the whole read much easier.

5. Regarding results: they are okay but it might make sense to provide more background about the data and also include links where we can find the data.

6. I would like to see some evaluations with other prior work. Otherwise, it is difficult to understand the superiority of your method compared to prior work.

7. There is no actual discussion about the results. Why the results are like this? How each different data affect the results? What kind of data would increase even more the performance or the accuracy of the results? 

8. The conclusions are not enough. I would like to read at least a paragraph regarding your future direction and how such a method could be implemented and tested in real-world.

9. Missing reference. You briefly mentioned the basics of AI in Section 1, there are various and important papers that should be included regarding ML/AI/Reinforcement Learning and so on. I am proving a list below.

-- Ngiam, J., Khosla, A., Kim, M., Nam, J., Lee, H., & Ng, A. Y. (2011). Multimodal deep learning. In Proceedings of the 28th international conference on machine learning (ICML-11) (pp. 689-696).

-- Mousas, C., & Anagnostopoulos, C. N. (2017). Learning Motion Features for Example-Based Finger Motion Estimation for Virtual Characters. 3D Research, 8(3), 25.

-- Nam, J., Herrera, J., Slaney, M., & Smith, J. O. (2012, October). Learning Sparse Feature Representations for Music Annotation and Retrieval. In ISMIR (pp. 565-570).

-- Rachmadi, M. F., Valdés-Hernández, M. D. C., Agan, M. L. F., Di Perri, C., Komura, T., & Alzheimer's Disease Neuroimaging Initiative. (2018). Segmentation of white matter hyperintensities using convolutional neural networks with global spatial information in routine clinical brain MRI with none or mild vascular pathology. Computerized Medical Imaging and Graphics, 66, 28-43.

-- Haarnoja, T., Zhou, A., Abbeel, P., & Levine, S. (2018). Soft actor-critic: Off-policy maximum entropy deep reinforcement learning with a stochastic actor. arXiv preprint arXiv:1801.01290.

--Haarnoja, T., Pong, V., Zhou, A., Dalal, M., Abbeel, P., & Levine, S. (2018, May). Composable deep reinforcement learning for robotic manipulation. In 2018 IEEE International Conference on Robotics and Automation (ICRA) (pp. 6244-6251). IEEE.

-- Krizhevsky, A., Sutskever, I., & Hinton, G. E. (2012). Imagenet classification with deep convolutional neural networks. In Advances in neural information processing systems (pp. 1097-1105).

-- Holden, D., Komura, T., & Saito, J. (2017). Phase-functioned neural networks for character control. ACM Transactions on Graphics (TOG), 36(4), 42.

-- Gao, Y., Lin, J., Yu, F., Levine, S., & Darrell, T. (2018). Reinforcement learning from imperfect demonstrations. arXiv preprint arXiv:1802.05313.

-- Mousas, C., Newbury, P., & Anagnostopoulos, C. N. (2014, May). Evaluating the covariance matrix constraints for data-driven statistical human motion reconstruction. In Proceedings of the 30th Spring Conference on Computer Graphics (pp. 99-106). ACM.

-- Suk, H. I., Wee, C. Y., Lee, S. W., & Shen, D. (2016). State-space model with deep learning for functional dynamics estimation in resting-state fMRI. NeuroImage, 129, 292-307.

-- Abdel-Hamid, O., Mohamed, A. R., Jiang, H., & Penn, G. (2012, March). Applying convolutional neural networks concepts to hybrid NN-HMM model for speech recognition. In Acoustics, Speech and Signal Processing (ICASSP), 2012 IEEE International Conference on (pp. 4277-4280). IEEE.

-- Chéron, G., Laptev, I., & Schmid, C. (2015). P-cnn: Pose-based cnn features for action recognition. In Proceedings of the IEEE international conference on computer vision (pp. 3218-3226).

-- Zimo Li, Y. Z., Xiao, S., He, C., & Li, H. (2017). Auto-Conditioned LSTM Network for Extended Complex Human Motion Synthesis. arXiv preprint. arXiv, 1707.

-- Mousas, C., Newbury, P., & Anagnostopoulos, C. N. (2014, September). Data-driven motion reconstruction using local regression models. In IFIP International Conference on Artificial Intelligence Applications and Innovations (pp. 364-374). Springer, Berlin, Heidelberg.

-- Saito, S., Wei, L., Hu, L., Nagano, K., & Li, H. (2017, July). Photorealistic facial texture inference using deep neural networks. In IEEE Conference on Computer Vision and Pattern Recognition, CVPR (Vol. 3).

-- Gu, S., Lillicrap, T., Sutskever, I., & Levine, S. (2016, June). Continuous deep q-learning with model-based acceleration. In International Conference on Machine Learning (pp. 2829-2838).

-- Bilmes, J. A., & Bartels, C. (2005). Graphical model architectures for speech recognition. IEEE signal processing magazine, 22(5), 89-100.

-- Rekabdar, B., Mousas, C., & Gupta, B. (2019, January). Generative Adversarial Network with Policy Gradient for Text Summarization. In 2019 IEEE 13th International Conference on Semantic Computing (ICSC) (pp. 204-207). IEEE.

-- Gu, S., Holly, E., Lillicrap, T., & Levine, S. (2017, May). Deep reinforcement learning for robotic manipulation with asynchronous off-policy updates. In 2017 IEEE international conference on robotics and automation (ICRA) (pp. 3389-3396). IEEE.

-- Li, R., Si, D., Zeng, T., Ji, S., & He, J. (2016, December). Deep convolutional neural networks for detecting secondary structures in protein density maps from cryo-electron microscopy. In Bioinformatics and Biomedicine (BIBM), 2016 IEEE International Conference on (pp. 41-46). IEEE. I believe that after a major revision, the paper will be ready for publication. 

-- Gu, S., Lillicrap, T., Ghahramani, Z., Turner, R. E., & Levine, S. (2016). Q-prop: Sample-efficient policy gradient with an off-policy critic. arXiv p

-- Rekabdar, B., & Mousas, C. (2018, November). Dilated Convolutional Neural Network for Predicting Driver's Activity. In 2018 21st International Conference on Intelligent Transportation Systems (ITSC) (pp. 3245-3250). IEEE.

In conclusion, overall, this paper has some research contributions, however, at this point, I recommend major revision. I feel that after addressing the mentioned issues the paper will be ready for publication.

Author Response

Dear professor,

First of all, thank you for your constructive comments on our manuscript. We have carefully revised our manuscript according to your comments, and the details is shown in what follows.

Point 1: Introduction: It is too short and it does not even introduce the problem and the method that was followed to solve the problem. Please consider revising this section. 

Response 1: We will revise the "Introduction" section and describe the problem and the method that is to be adopted to solve the problem in detail, and more literature are reviewed in the added "literature review" section.

Point 2: There is no Literature Review section in this paper (or there is. some lit review in the Introduction section), which in most cases is Section 2. Instead, Section 2 describes the HFSP. I would recommend including a literature review section with prior work that has been conducted towards traffic light period configuration.

Response 2: We elaborate the section of “Literature review” in the revised manuscript in which lots of prior works are reviewed.

Point 3: Current Section 2 should become Section 3 and so on...

Response 3: We will remark the Section after adding the Section “literature review”.

Point 4: Section 3 is quite informative but I would like to see a table with notations at the beginning. This would make the whole read much easier.

Response 4: At the beginning of the 3 Section, a summary description is added such as Section 3.1 describe ... Section 3.2 analyse…and so on. So the whole will be easier to read.

Point 5: Regarding results: they are okay but it might make sense to provide more background about the data and also include links where we can find the data.

Response 5: We provide the source of the data in the revised manuscript and analyse the reason why the data is like this.

Point 6: I would like to see some evaluations with other prior work. Otherwise, it is difficult to understand the superiority of your method compared to prior work.

Response 6: In our revised manuscript, we add the section “analysis of complexity” where we analyse the complexity of the genetic algorithm (GA) and the artificial immune system (AIS) and RL. We evaluate GA, AIS and RL from complexity, real-time performance and other respects.

`

Point 7: There is no actual discussion about the results. Why the results are like this? How each different data affect the results? What kind of data would increase even more the performance or the accuracy of the results?  

Response 7: We elaborate the paragraph of “discussion” in which we explain why the results are like this. Besides, we analyse the effect of the relative parameters on the results, for instance, why the corresponding parameters value  \omega and b are chosen and how they influence the result.

Point 8: The conclusions are not enough. I would like to read at least a paragraph regarding your future direction and how such a method could be implemented and tested in real-world.

Response 8: To test the RL method, we can compare the manual scheduling and the scheduling obtained by other algorithms. In the section “Conclusion”, we describe our future direction, i.e. constructing an repository of sortie scheduling of carrier aircraft in continuous dispatch with different the number of carrier aircraft using the RL method, by which our method can provide some instruction to people in relative field such as the Carrier Commanders.

Point 9: Missing reference. You briefly mentioned the basics of AI in Section 1, there are various and important papers that should be included regarding ML/AI/Reinforcement Learning and so on. I am proving a list below.

Response 9: We expand our section “Introduction” and add the section “literature review” where an array of literatures are reviewed detailedly, especially these you recommend are included.

Reviewer 2 Report

The authors proposed to apply reinforcement learning (RL) to solve hybrid flow shop scheduling problems (HFSP). The authors adopted the Boltzmann exploration policy in action selection. The authors investigated the performance of the proposed method using two case studies and concluded that RL outperforms GA. The novelty of the work is not high. Application of RL on flow shop scheduling problems has been widely studied. It is a straightforward extension to apply RL to hybrid flow shop scheduling problem. However, the authors adopted the Boltzmann exploration policy for action selection, which common in RL community. The review on different approaches for finding solutions for HFSP is very brief. The authors should elaborate on this review section. The differences between traditional and hybrid flow shop scheduling problems are not clearly stated. The challenges in finding HFSP optimal solution are not highlighted. The definition of state in the RL formulation of HFSP is not clear. Should the state be (Stage, workpiece) instead of (state,workpiece) in line119. The rationale of the (linear) form of the proposed reward function in equations (17) and (18) was not clearly stated. what are the effect of \omega and b on the performance of algorithm? The authors should justify the choices of parameters in the RL  in lines 201-202 and discuss the effects of these parameters to the performance (accuracy and speed) of the propsoed algorithm. The authors are suggested that they discuss the complexity of AIS, GA and the proposed RL method. In Table 2, the labels of the machines are not consistent with those in Table 1. In line 218, define a stage as in "... of the third stage." Does it refer to one of the types of machines: lathing, planing and grinding? 

Author Response

Dear professor,

First of all, thank you for your constructive comments on our manuscript. We have carefully revised our manuscript according to your comments, and the details is shown in what follows.

Point 1: The review on different approaches for finding solutions for HFSP is very brief. The authors should elaborate on this review section. 

Response 1: We elaborate the section of “Literature review” in which we comment an array of literatures involving different approaches for finding solutions for HFSP.

Point 2: The differences between traditional and hybrid flow shop scheduling problems are not clearly stated. The challenges in finding HFSP optimal solution are not highlighted.

Response 2: In section “literature review”, we distinguish the traditional flow shop scheduling problems with the hybrid one. And we describe the challenges in finding HFSP optimal solutions as NP-hard problem.

Point 3: The definition of state in the RL formulation of HFSP is not clear. Should the state be (Stage, workpiece) instead of (state,workpiece) in line119.

Response 3: We are ashamed of making such mistake, and we correct it as soon as we have seen your comment.

Point 4: The rationale of the (linear) form of the proposed reward function in equations (17) and (18) was not clearly stated. what are the effect of \omega and b on the performance of algorithm? The authors should justify the choices of parameters in the RL in lines 201-202 and discuss the effects of these parameters to the performance (accuracy and speed) of the propsoed algorithm.

Response 4: We add more descriptions for equations (17) and (18), explaining the reasonability of equation. In section “case validation”. We discuss the impact of parameter  and on the performance (quality of optimal solution) of the algorithm, and we justify the choice of parameters when adopting equation (17).

Point 5: The authors are suggested that they discuss the complexity of AIS, GA and the proposed RL method.

Response 5: We add the section “analysis of complexity” under the section “case validation” to analyse the complexity of RL and compare the complexity of genetic algorithm (GA), artificial immune system (AIS) and RL.

Point 6: In Table 2, the labels of the machines are not consistent with those in Table 1. In line 218, define a stage as in "... of the third stage." Does it refer to one of the types of machines: lathing, planing and grinding?

Response 6: In Table 2, the labels in “Times row” is the times of computing, not the labels of the machines. The potential fault is the tags of workpieces in Table 1 not consistent with Figure 3, which has been corrected. Besides, in line 218 , “the third stage” does mean “grinding” stage and we change the expression as “the third stage or grinding stage”.

Reviewer 3 Report

Referee’s report on the manuscript:

 “A Reinforcement Learning Method for Hybrid Flow-shop Scheduling Problem”

Submitted for the publication on the MDPI-Algorithms Journal

 PAPER’S ID:  ALGORITHMS-610524

 GENERAL COMMENTS:

 Authors consider a relevant problem (i.e., Hybrid Flow-shop Scheduling Problem) in the field of scheduling, and present an interesting solution approach based on a reinforcement learning method. Authors illustrate the behavior of the proposed methodology conducting a set of examples on selected case studies.

 The manuscript discusses on a methodology which can be of potential interest also in different contexts. After a description of the specific scheduling problem, the proposed approach and a review of the relevant literature, authors illustrate their ideas showing their characteristics and benefits on the basis of some examples.

The scientific work reported in the submitted manuscript is of clear interest under both the research and the application point of view.

 Overall, I believe that the paper presents an interesting study for a relevant application in the field of scheduling. The topic of the paper seems to be sound with respect to MDPI-Algorithms journal, but some aspects of the manuscript need to be improved before publication can be considered.

 In order to improve the overall quality of the paper, there are more detailed and specific suggestions and remarks reported in what follows.

Other comments and remarks:

The first time authors use an initialism or acronym in their manuscript, the words should be written (please check) out with the short form placed in parentheses. This way, it's clear to the readers exactly what the letters mean. The quality and readability of the figures should be improved (pictures, data and text) and more explicative caption and descriptions should be used. In the abstract (and in the paper as well) it is not clear how a FCFS rule can be defined as "intelligent algorithm". Moreover, in the ending sentence of the abstract the meaning of “practical” needs to be clarified. In their introduction or in the successive literature review and references, authors should also consider (or discuss) also the possibility to use “rollout” algorithmic schemes, and some papers could be considered. For instance I found the following works:

-   M Ciavotta, C Meloni, M Pranzo, Speeding up a Rollout algorithm for complex parallel machine scheduling, International Journal of Production Research, 2016, 54, 16, 4993-5009.

-   Ciavotta M., Meloni C., Pranzo M., Scheduling dispensing and counting in secondary pharmaceutical manufacturing. AIChE Journal, vol. 55 (5), pp. 1161-1170, 2009.

In general, the paper needs to be carefully formatted according to the journal editorial rules. Page 5 and 6: Algorithm 1 should appear in one single page. The values of all parameters and coefficients used in the experiments need some motivation. How are obtained these values? Is the behavior of the method affected by these choices? In Section 4 , lines 203-209 are not clear and needs a better explanation. Moreover, the computation effort (e.g., time) required by the algorithms should be reported and discussed. The computing environment should be introduced in a more clear (i.e., giving its technical characteristics) way. E.g., The Anaconda version is too vague as description. Instead, used hardware and the specific software is of interest here. As the manuscript discusses on a methodology which is of potential interest also in different contexts. Authors should illustrate –at least in the conclusion- their ideas to extend/generalize this approach on the basis of its characteristics and benefits. Moreover, the consideration about “real-time” applications needs to be addressed more in detail in the paper discussing the computation time required by the algorithms (see also other comments). From the presentation point of view, even if the form is adequate, it is suggested that an additional proof reading be done by the authors before submitting a new version of the paper. The list of references needs to be formatted following the journal rules.

Author Response

First of all, thank you for your constructive comments on our manuscript. We have carefully revised our manuscript according to your opinions, and the details is shown in what follows.

Point 1: The first time authors use an initialism or acronym in their manuscript, the words should be written (please check) out with the short form placed in parentheses. This way, it's clear to the readers exactly what the letters mean.

Response 1: We write out the words before the acronym AIS and regulate the acronym RL

Point 2: The quality and readability of the figures should be improved (pictures, data and text) and more explicative caption and descriptions should be used.

Response 2: We improve the quality as well as possible and change the annotation, data, text of all figures to reinforce the readability. Besides, we change the captions of some figure and add some necessary to make it clear for readers to understand what the figures reveal .

Point 3: In the abstract (and in the paper as well) it is not clear how a FCFS rule can be defined as "intelligent algorithm".

Response 3: This should be our wrong expression, we do not have the viewpoint that a FCFS rule can be defined as "intelligent algorithm". To correct this, we alter the description in the abstract to "Compared with the first-come-first-serve strategy which is frequently adopted when coding in most of traditional intelligent algorithms".

Point 4: In the ending sentence of the abstract the meaning of “practical” needs to be clarified.

Response 4: We analysis the complexity of genetic algorithm (GA), artificial immune system (AIS) and reinforcement learning (RL) in our manuscript. By changing the description in the ending of abstract to " The results demonstrate that the machining and support scheduling obtained by this RL method is reasonable in quality, real-time performance and complexity, indicating that this RL method is practical for HFSP. ", we clarify the practicality of the RL for HFSP.

Point 5: In their introduction or in the successive literature review and references, authors should also consider (or discuss) also the possibility to use “rollout” algorithmic schemes, and some papers could be considered.

Response 2: After reading the paper you recommend, we decode to add the section "Literature review", in which we consider the “rollout” algorithmic schemes and the literatures you recommend is especially included.

Round 2

Reviewer 1 Report

After carefully reading the revised version of the paper as well as the responses made by the authors of this paper, I feel confident that this is a strong and scientifically sound paper. For this reason, I would like to recommend this paper for the Algorithms Journal. Well done!

Reviewer 2 Report

The authors have successfully addressed my comments and thus I recommend the acceptance of the manuscript in the journal Algorithms.

Reviewer 3 Report

Authors satisfactorily addressed all my comments and remarks.